# A Novel Synergistic Flame Retardant of Hexaphenoxycyclotriphosphazene for Epoxy Resin

**DOI:** 10.3390/polym13213648

**Published:** 2021-10-23

**Authors:** Jiawei Jiang, Siqi Huo, Yi Zheng, Chengyun Yang, Hongqiang Yan, Shiya Ran, Zhengping Fang

**Affiliations:** 1Laboratory of Polymer Materials and Engineering, NingboTech University, Ningbo 315100, China; 714413862@163.com (J.J.); sqhuo@nbt.edu.cn (S.H.); zy13357018173zy@163.com (Y.Z.); yangchengyun991007@163.com (C.Y.); zpfang@nbt.edu.cn (Z.F.); 2College of Chemical and Biological Engineering, Zhejiang University, Hangzhou 310027, China

**Keywords:** epoxy resin, Hexaphenoxycyclotriphosphazene, metal-organic frame, halloysite nanotubes, flame retardancy, mechanical properties

## Abstract

Hexaphenoxycyclotriphosphazene (HPCP) is a common flame retardant for epoxy resin (EP). To improve the thermostability and fire safety of HPCP-containing EP, we combined UiO66-NH_2_ (a kind of metal-organic frame, MOF) with halloysite nanotubes (HNTs) by hydrothermal reaction to create a novel synergistic flame retardant (H-U) of HPCP for EP. For the EP containing HPCP and H-U, the initial decomposition temperature (*T*_5%_) and the temperature of maximum decomposition rate (*T*_max_) increased by 11 and 17 °C under nitrogen atmosphere compared with those of the EP containing only HPCP. Meanwhile, the EP containing HPCP and H-U exhibited better tensile and flexural properties due to the addition of rigid nanoparticles. Notably, the EP containing HPCP and H-U reached a V-0 rating in UL-94 test and a limited oxygen index (LOI) of 35.2%. However, with the introduction of H-U, the flame retardant performances of EP composites were weakened in the cone calorimeter test, which was probably due to the decreased height of intumescent residual char.

## 1. Introduction

Epoxy resin (EP), as a crucial thermoset polymer, and is extensively used in various industries for its excellent mechanical properties, electrical insulation, adhesion strength and chemical resistances [1,2,3,4,5]. Nevertheless, EP is inherently flammable due to its chemical constitution, which may bring hidden danger to people’s life and property. Once ignited, EP burns vigorously, and a lot of heat and smoke will be released. Thus, it is of great significance to improve its flame retardancy [6,7,8]. Traditionally, halogen-based flame retardants (FRs) have been widely applied for their high efficiency in improving the flame retardancy of EP. However, EP with halogen-based, especially brominated FRs, may produce poisonous smoke and toxic halogenated dibenzodioxins and dibenzofurans during combustion [9,10]. In consideration of environmental problems, halogen-free FRs have consequently attracted a lot of attention [11,12,13]. Among all halogen-free FRs, phosphorus-containing FRs are considered as a promising candidate for EP due to their low toxicity and high efficiency [14,15].

Among different kinds of phosphorous-containing FRs, cyclotriphousphazene and its derivatives have been widely used in flame-retardant EP [16,17]. When used alone, they usually promote the formation of an intumescent char layer during combustion. However, the intumescent char layer is not compact enough, which may restrain further improvement in flame retardant performance [18]. Halloysite nanotubes (HNTs, Al_2_(OH)_4_Si_2_O_5_⋅nH_2_O) are one class of common clays with a unique hollow tubular structure in nature [19]. HNT is characterized by its rich surface pore structure, high adsorption properties, large aspect ratio, and outstanding mechanical properties and thermal stability, and thus it has been widely used in drug carriers, nanocomposites, and biosensors [20,21,22,23]. Additionally, HNT is also applied as a flame retardant for different polymeric materials. As a rigid and thermally stable nanoadditive, HNT can not only improve the mechanical strength and modulus of the composite materials, but can also greatly enhance the thermal stability and wear resistance [24,25]. However, HNT is often combined with other FRs, because HNT only slows down the combustion process and cannot reduce the amount of combustible materials. Therefore, it is better to use HNT as a synergistic FR of P-containing FRs [26]. Metal-organic framework (MOF) is a porous crystalline polymer formed by coordination bonds between organic ligands and metal ions or ion clusters, which has the advantages of high porosity, large specific surface area, easy functionalization and modification [27]. MOF has made good progress in the fields of biomedicine, electrode materials, gas storage and separation, showing great application potential [28,29,30,31]. Meanwhile, MOF has also been used as a flame retardant for different polymeric materials. For example, Sai et al. reported that the porous structure of MOF can delay the time to ignition while the flame retardant efficiency improve due to the charring or crosslinking reaction catalyzed by metal element [32]. Hou et al. reported that the combination of the adsorption and catalytic effect of P-MOF enhanced the fire safety of EP [33]. Zhang et al. modified MOF(UiO66-NH_2_) with phytic acid, which led to the reinforced char with a strong barrier and a higher polyaromatic structure for EP samples [34].

In this paper, we developed a novel flame-retardant nanomaterial (H-U) by combining HNT with UiO66-NH_2_, which was used as a synergistic flame retardant for hexaphenoxycyclotriphosphazene (HPCP) in epoxy resin. The effects of H-U and HPCP on the thermal stability, mechanical properties and flame retardancy of EP were studied in detail. Additionally, the flame retardant mechanism was studied by different tests.

## 2. Materials and Methods

### 2.1. Materials

Diglycidyl ether of bisphenol A (DGEBA) with an epoxy value of 0.53 mol/100 g was provided by Yueyang Baling Huaxing Petrochemical Co., Ltd. (Hunan, China). Halloysite was purchased from Runwo Materials Technology Co., Ltd. (Guangzhou, China). 2-Aminoterephthalic Acid (BDC, 95%) and zirconium (IV) tetrachloride (ZrCl_4_, 98%) were purchased from Sigma-Aldrich (St. Louis, MO, USA). Trichloromethane (CHCl_3_), toluene, methanol, ethanol and *N, N*-dimethylformamide (DMF) were provided by Sinopharm Chemical Reagent. Co. Ltd. (Shanghai, China). 3-Aminopropyltriethoxy silane (APTES) and trimethylamine (TEA) were purchased from Sigma-Aldrich (St. Louis, MO, USA). Hexaphenoxycyclotriphosphazene (HPCP, 98%) was purchased from Macklin Biochemical Co., Ltd. (Shanghai, China).

### 2.2. Preparation of HNTs-UiO66 (H-U)

There are plentiful active groups on the surface of HNTs, and thus different kinds of functional groups can be introduced. In this work, the carboxyl functionalized HNTs (HNTs-COOH) were synthesized based on previous research [35]. Firstly, in a 500 mL single-necked flask, 10 g HNTs were uniformly dispersed in 250 mL of toluene through ultrasonic dispersion and magnetic stirring. Then, 10 mL of TEA and 20 mL of APTES were added to the flask, and the mixture was stirred for 24 h at 80 °C under nitrogen atmosphere. The APTES-modified HNTs (HNTs-NH_2_) were centrifuged and washed with deionized water and ethanol. In addition, then, HNTs-NH_2_ was placed in a vacuum oven at 50 °C for 24 h. After that, 5 g HNTs-NH_2_ powder and 2 g succinic anhydride were added into 200 mL DMF, and the mixture was stirred at 25 °C for 24 h. Similarly, the obtained HNTs-COOH was washed with deionized water and ethanol, and the washed product was dried at 50 °C under vacuum for 24 h. Afterwards, based on our previous work [36], 3.50 g ZrCl_4_ and 2.49 g BDC-NH_2_ were dissolved in 210 mL DMF with magnetic stirring for 30 min, and then 1 g HNTs-COOH was added into mixture with continuous stirring for 15 min. The obtained mixture was transferred to a high-pressure autoclave, and placed at 120 °C in an oven for 24 h. The obtained product was centrifuged, and washed with DMF, CHCl_3_ and methanol for several times, and finally dried at 120 °C under vacuum for 24 h. The synthetic route of H−U is shown in Figure 1.

### 2.3. Fabrication of EP Thermosets

The fabrication process of EP thermosets was as follows. A certain amount of HPCP and H-U were added to a 250 mL three-necked flask, and then 30 mL acetone was added, followed by ultrasonically dispersing for 15 min. A certain amount of epoxy resin was added to the flask and then the mixture was mechanically stirred at 120 °C for 2 h. After that, 4, 4’-diaminodiphenylsulfone (DDS) was added to the mixture, and stirred for 15 min. The obtained mixture was heated to 130 °C and defoamed under vacuum for 5 min. After the bubbles were removed, the mixture was poured into a mold that was preheated at 130 °C in an oven. The mixture was cured at 130 °C for 2 h, 180 °C for 5 h and 200 °C for 2 h, respectively. Finally, EP thermosets were obtained after naturally cooled to room temperature. The formulas of epoxy thermosets are listed in Table 1.

### 2.4. Characterization and Measurements

The Fourier transform infrared spectra were obtained using a Vector-22 FTIR spectrophotometer (IR, Bruker, Karlsruhe, Germany). Thermal gravimetric analysis (TGA) was carried out in N_2_ and air conditions at a heating rate of 20 °C/min from 30 to 800 °C using a TGA analyzer (209 F1, Netzsch, selb, Germany). Differential scanning calorimetry (DSC) was performed on a PerkinElmer DSC 4000 (PerkinElmer, Waltham, MA, USA) at a heating rate of 10 °C/min under nitrogen. Dynamic mechanical analysis (DMA) was carried out on a DMA Q800 apparatus (TA Instruments, New Castle, DE, USA) under single cantilever bending mode at a heat-up rate of 3.00 °C/min from 30 to 250 °C. Mechanical properties were evaluated by a CMT6104 universal testing machine (MTS Systems Co., Ltd., Nanjing, China). The tensile test was conducted based on ASTM D638 by using the dumbbell-shaped samples. Three-point flexural measurements were undertaken in accordance with ASTM D790. Five specimens for each sample were used, and the average values were reported. Limiting oxide index (LOI) was recorded on a LOI tester (HC-2, Jiangning Analyzer Instrument, Nanjing, China) according to GB2406-80, and the dimension of samples was 130 mm × 6.5 mm × 3 mm. UL-94 vertical burning tests were conducted using a vertical burning instrument (CZF-3, Jiangning Analyzer Instrument, Nanjing, China) with specimen dimensions of 130 mm × 13 mm × 3 mm according to ASTM D3801. Cone calorimeter tests were performed on a cone calorimeter (CONE, Fire Testing Technology, East Grinstead, UK) according to ISO-5660. Square specimens (100 mm × 100 mm × 3 mm) were irradiated at a heat flux of 35 kW/m^2^. Typically, three specimens were needed for each sample, and the error of the obtained data was reproducible within ±5%. The morphology of residual char after cone calorimeter test was observed by a scanning electron microscope (SEM, S-4800, Hitachi, Japan). A transmission electron microscope (TEM, JEOL, 1230, Akishima, Japan) was used to acquire detect the phase morphology of epoxy composites. Thermogravimetric analysis/infrared spectrometry (TGA-FTIR) was conducted using a TGA analyzer coupl ed with a Thermo Nicolet IS10 FTIR spectroscopy (TGA, 209 F1, Netzsch, selb, Germany; Nicolet IS10 FTIR, Madison, Wisconsin, USA) under N_2_ conditions at a heating rate of 20 °C/min from 30 to 800 °C. About 6.00 mg sample was used.

## 3. Results and Discussion

### 3.1. Characterization of H-U

TEM images of HNTs, HNTs-COOH, UiO66-NH_2_, and H-U, and SEM images of HNTs-COOH and H-U are presented at Figure 2. As shown in Figure 2a, the HNTs possessed a unique hollow tubular structure with large length-diameter ratio. When modified with -COOH, the surface of HNTs became rough, and the diameter was larger than that of the unmodified HNTs. Unlike HNTs, the UiO-66-NH_2_ was an irregular sphere (see Figure 2d). For the H-U in Figure 2f, the surface of the tube was covered with a layer of irregular spheres, which illustrated that UiO66-NH_2_ had been successfully wrapped onto the surface of the HNTs of H-U tube.

FT-IR technique was also employed to characterize the structure change of nanomaterials, with the spectra shown in Figure 3. The absorption peaks at 3695 and 3623 cm^−1^ belonged to the stretching vibration of the hydroxyl groups of HNTs. The sharp absorption peak at 913 cm^−1^ was attributed to the hydroxyl group attached to the aluminum. Additionally, there was an absorption peak of Si-O bond at 1033 cm^−1^. For HNT-NH_2_, the stretching peaks of -NH_2_ groups appeared around 3451 cm^−1^. The carbonyl vibration peaks of HNTs-COOH could be observed at 1542 and 1646 cm^−1^, respectively. For UiO66-NH_2_, the broad band between 3374 and 3470 cm^−1^ demonstrated the presence of the -NH_2_ groups. The absorption peaks around 1256 and 1435 cm^−1^ were assigned to the absorption peaks of C-N and N-H groups. The peaks located at 1388 and 1560 cm^−1^ were assigned to the asymmetric and symmetric stretching vibrations of the -COO- connected with Zr^4+^, respectively. Generally, H-U showed similar absorption peaks to UiO66-NH_2_. However, an absorption peak of Si-O bond appeared at 1033 cm^−1^ in the FT-IR spectrum of H-U, further indicating that UiO66-NH_2_ had been successfully grafted on the surface of HNTs.

### 3.2. Thermal Properties

The glass transition temperature (*T*_g_) of the cured epoxy resin is important information for evaluating its thermal resistance, and thus DSC was applied to investigate the influence of H-U on the *T*_g_ of EP samples. As shown in Figure 4, the *T*_g_ value of pure EP/DDS was as high as 195 °C. With the introduction of 9.0 wt% HPCP, the *T*_g_ value of EP/DDS/HPCP-9 obviously reduced to 185 °C due to the plasticizing effect of HPCP. Notably, replacing part of HPCP with H-U contributed to increasing the *T*_g_ values of EP/DDS/HPCP/H-U samples. For instance, the EP/DDS/HPCP-6/H-U-3 sample exhibited a *T*_g_ value of 192 °C, which was very close to that of EP/DDS sample. It was supposed that the incorporation of rigid H-U restricted the mobility of epoxy chains and thus increased the *T*_g_ value of EP samples.

Additionally, TGA test was also performed to further estimate the thermal stability of the cured epoxy resins. The TG and DTG curves of epoxy thermosets are shown in Figure 5. The initial decomposition temperature (temperature at 5% weight loss, *T*_5%_), temperature of maximum decomposition rate (*T*_max_), maximum decomposition rate (R_max_) and char residue at 800 °C (CY) are listed in Table 2.

In N_2_ conditions, the *T*_5%_ and *T*_max_ values of pure EP/DDS reached 391 °C and 433 °C, respectively. The addition of HPCP reduced the *T*_5%_ and *T*_max_ values of EP/DDS/HPCP-9 sample due to the promoting decomposition effect of phosphorus-based compounds from HPCP [16,37]. With the increasing loading level of H-U, the *T*_5%_ values of the cured epoxy resins gradually increased, indicating that the thermal stability was effectively remedied. This can be explained as follows: firstly, the increase in *T*_5%_ can be primarily attributed to the decreased content of phosphorus; secondly, the interfacial reaction between HNTs and EP matrix can also retard the decomposition at initial stage [38]. At high temperature, inorganic particles can hinder the movement of polymer chain in the interface region between polymer and inorganic phase [39]. Meanwhile, the presence of thermally stable UiO66-NH_2_ particles can also improve the thermal resistance of EP composites [40]. For *T*_max_, it showed a similar trend to the initial decomposition temperature. Additionally, as shown in Table 2, the char yield of EP composite was obviously increased when 9 wt% HPCP was added, indicating that HPCP effectively promoted the carbon formation of epoxy matrix. After replacing 1 wt% HPCP with H-U, the char yield increased from 20.0% to 21.5%. However, the further increase in H-U content decreased the char yield of EP thermosets, and the char yield of EP/DDS/HPCP-6/H-U-3 sample was basically consistent with that of EP/DDS/HPCP-9 sample.

In air conditions, the cured epoxy resins showed two degradation stages. The first stage was due to the degradation of principal polymer networks, while the second one was attributed to the oxidative degradation of formed char [41,42]. In the initial thermal oxidation degradation stage, the presence of H-U promoted the thermal oxidative stability of EP/DDS/HPCP composite. At elevated temperature, H-U also increased the char yield of EP/DDS/HPCP/H-U composites.

### 3.3. Mechanical Properties

The DMA test was conducted to study the dynamic mechanical properties of the cured epoxy resins, and the curves of storage modulus (*E*′) and tan delta (Tan*δ*) versus temperature are shown in Figure 6. Storage modulus is an important parameter to evaluate the rigidity of materials. As shown in Figure 6, the *E*′ of epoxy thermoset at 30 °C was significantly reduced after the addition of HPCP. With the addition of H-U, the *E*′ at 30 °C was increased. For instance, the *E*′ of EP/DDS/HPCP-6/H-U-3 sample reached 2652 MPa, which was little higher than that of pure EP/DDS sample. It was inferred that HNTs, as a rigid nanotube, increased the rigidity of EP thermoset through the interface reaction between HNTs and matrix [39]. Meanwhile, the amine groups on the surface of H-U might also react with the epoxide groups during curing process, thus promoting the interfacial strength [34]. In addition, the temperature corresponding to the peak value of the tan delta curve is the glass transition temperature (*T*_g_). Similar to *E*′, the *T*_g_ value of EP/DDS/HPCP-9 sample was lower than that of EP/DDS sample. With the addition of H-U, the *T*_g_ was almost unchanged, which may be assigned to two competitive factors: the rigid-phase reinforcement and destroying of the epoxy network structure [43,44,45].

Additionally, the tensile and flexural properties of EP thermosets were also tested to further study the effect of H-U on mechanical properties of the cured epoxy resins, and the data is shown in Table 3. Apparently, the tensile and flexural strengths of EP thermosets were reduced when adding 9 wt% HPCP, indicating the negative effect of HPCP on the mechanical properties of EP thermosets. However, when a part of HPCP was replaced by H-U, the mechanical properties were improved. Especially for EP/DDS/HPCP-7/H-U-2 sample, the tensile and flexural strengths were 68.75 MPa and 108.01 MPa, which were fairly close to those of EP/DDS sample. On the other hand, the elastic and flexural moduli were continuously increased with the increasing content of H-U. In this study, UiO66-NH_2_ was covalently bonded with HNTs to improve the compatibility between EP and nanoparticles [46]. As shown in Figure 7, H-U was uniformly dispersed in the epoxy matrix. As a result, EP/DDS/HPCP/H-U samples exhibited better mechanical properties in comparison with EP/DDS/HPCP-9 sample, indicating that H-U, as a synergist of HPCP, owned its practical significance.

### 3.4. Flame Retardant Performance

To study the synergistic flame retardant effect of HPCP and H-U, limiting oxygen index (LOI), UL-94 vertical burning and cone calorimeter tests were conducted. The corresponding data are presented in Table 4. Pure EP burned vigorously in air, and it suffered a low LOI value of 24.5% and could not pass any rating in the UL-94 vertical burning test. For the EP/DDS/HPCP-9 sample, it could not self-extinguish after the flame was removed in the UL-94 test, and thus its UL-94 rating was only NR. However, the LOI value of EP/DDS/HPCP-9 increased to 28.4%. This phenomenon can be assigned to the intumescent char layer immediately formed after ignition, which can suppress the combustion to some extent, but the layer was not dense enough to inhibit further combustion. With the increasing content of H-U, the LOI values and UL-94 ratings were significantly improved despite the decreasing content of HPCP. For EP/DDS/HPCP-6/H-U-3, its LOI reached up to 35.2%, UL-94 rating increased to V-0, and it showed good self-extinguishing performance, indicating the flame retardant effect of H-U. Digital images of EP/DDS/HPCP-9 and EP/DDS/HPCP−6/H−U−3 samples after UL-94 tests are shown in Figure 8. After the addition of H-U, the thermal resistance of EP nanocomposites was improved (as described in Section 3.2), and thus the nanocomposites were easier to self-extinguish after being ignited. Meanwhile, the denser residual char was formed with the introduction of H-U, which prevented the transfer of heat and mass. This may be the reason that EP/DDS/HPCP/H-U samples performed better than EP/DDS/HPCP-9 sample in LOI and UL-94 tests.

Additionally, a cone calorimeter test was also performed to assess the flammability of polymers. The experimental results are listed in Figure 9 and Table 4. Similar to TG results, the TTI of EP/DDS/HPCP-9 sample was reduced due to the catalytic effect of phosphorus-based compounds. With the addition of H-U, the TTIs of EP/DDS/HPCP/H-U samples increased. In addition, incorporating HPCP reduced the pHRR and THR from 942 kW/m^2^ and 84.4 MJ/m^2^ of EP/DDS sample to 481 kW/m^2^ and 41.7 MJ/m^2^, respectively. Such notable reductions may be due to the protection effect of intumescent char layer formed by HPCP under long-term thermal radiation. However, with the addition of H-U, the height of the char layer decreased significantly, as shown in Figure 10. The decrease in the height of the intumescent char layer resulted in the weakening of heat insulation and smoke suppression. Hence, the pHRR and THR of EP/DDS/HPCP/H-U samples increased to some extent. The phosphorus-based fragments produced by HPCP can promote the formation of stable intumescent char layer on the surface of epoxy matrix. Under the circumstance of constant heat, such stable intumescent layer can reduce the radiation of heat to the matrix, suppress the decomposition of the matrix and the release of flammable gases. As a result, the heat release was reduced during the combustion process. However, the thickness of the layer decreased with the introduction of H-U. Hence, the flame retardant effect of the char in the condensed phase was weakened. Likewise, the SPR and TSR values were also increased due to the decreased thickness of char layer. In sum, the introduction of H-U can enhance the performances of EP/DDS/HPCP sample in LOI and UL-94 tests due to its promotion effect in improving the char density, but under constant heat radiation, introducing H-U reduced the height of char, which weakened the flame retardancy and smoke suppression of EP samples.

### 3.5. Flame Retardant Mechanism

TG-IR under N_2_ conditions was conducted to detect the evolution of volatilized products for EP samples during thermal degradation. The TG-IR spectra of gas products of EP thermosets are shown in Figure 11 and Figure 12. As presented in Figure 11 and Figure 12, both EP/DDS/HPCP-9 and EP/DDS/HPCP-6/H-U-3 showed similar characteristic peaks at 3720–3860 cm^−1^ (O-H), 2020–2230 cm^−1^ (carbon monoxide), 1730 cm^−1^ (C-O), 1530 cm^−1^ (benzene), and 1160 cm^−1^ (aliphatic compounds). Figure 12a showed total absorbance of decomposition products of EP composites. Obviously, the total absorbance of decomposition products of EP/DDS reached a maximum while that of EP/DDS/HPCP-9 was on the lowest level. Additionally, the absorbance of aliphatic compounds at 1160 cm^−1^ and hydrocarbons at 2970 cm^−1^ of EP/DDS/HPCP-6/H-U-3 were higher than that of EP/DDS/HPCP-9. This phenomenon indicates that the introduction of H-U leads to a worse carbon promotion effect, which matches the previous test results.

The morphology and chemical composition of the char residues after cone calorimeter tests were analyzed to study the flame retardant effect in condensed phase by SEM. As presented in Figure 13a, the char residue of pure EP/DDS was loose, with lots of pore spaces, indicating that pure EP thermosets cannot form continuous and stable char during combustion. In terms of EP/DDS/HPCP-9, the char residue was much denser due to the introduction of P-containing flame retardant, which is conducive to heat and oxygen isolation and combustible gas suppression. Noticeably, for EP/DDS/HPCP-6/H-U-3, the char residue was compact and continuous. These results indicate that H-U can effectively promote the carbon formation of epoxy resin during combustion, thus endowing the epoxy resin with good flame retardancy. The EDS spectra showed that the carbon was the major component of char residues. In addition, a lot of P elements appeared in the char residue of EP/DDS/HPCP-9, indicating that the catalytic charring effect of P-containing groups in the condensed phase. For EP/DDS/HPCP-6/H-U-3, the char residues contained Zr (4.69%) and Al (1.11%) elements, which suggested that the presence of Al and Zr might be responsible for catalyzing the formation of denser char layers.

## 4. Conclusions

In this study, H-U was synthesized via hydrothermal reaction by combining UiO66-NH_2_ with HNT-COOH, which was used as a novel synergistic flame retardant of HPCP for EP. The introduction of H-U can effectively improve the thermostability, tensile and fluxural properties. Additionally, due to improved thermal resistance and the formation of a dense char, EP/DDS/HPCP-6/H-U-3 reached a V-0 rating in UL-94 test and achieved a limited oxygen index of 35.2%. However, the height of the char layer of the EP/DDS/HPCP/H-U samples was reduced due to the introduction of H-U, weakening their flame-retardant performances in the cone calorimeter test.

## Figures and Tables

**Figure 1 polymers-13-03648-f001:**
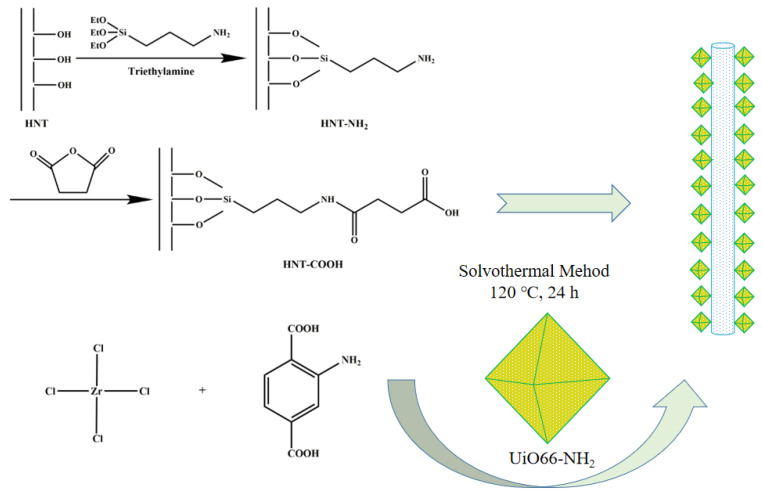
The synthesis of H-U.

**Figure 2 polymers-13-03648-f002:**
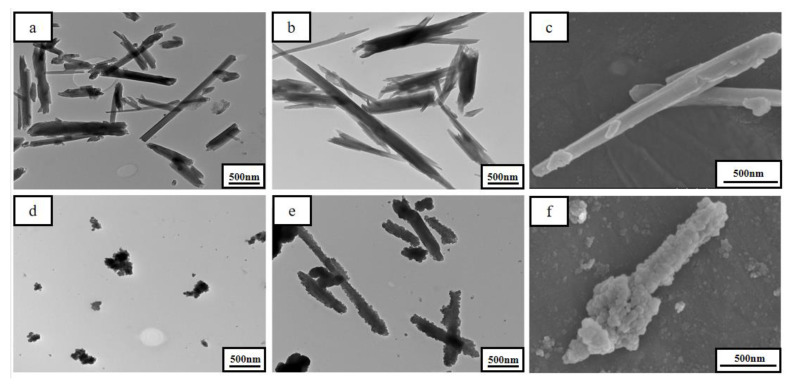
TEM images of HNTs (**a**), HNTs-COOH (**b**), UiO66-NH_2_ (**d**), and H-U (**e**); and SEM images of HNTs-COOH (**c**) and H-U (**f**).

**Figure 3 polymers-13-03648-f003:**
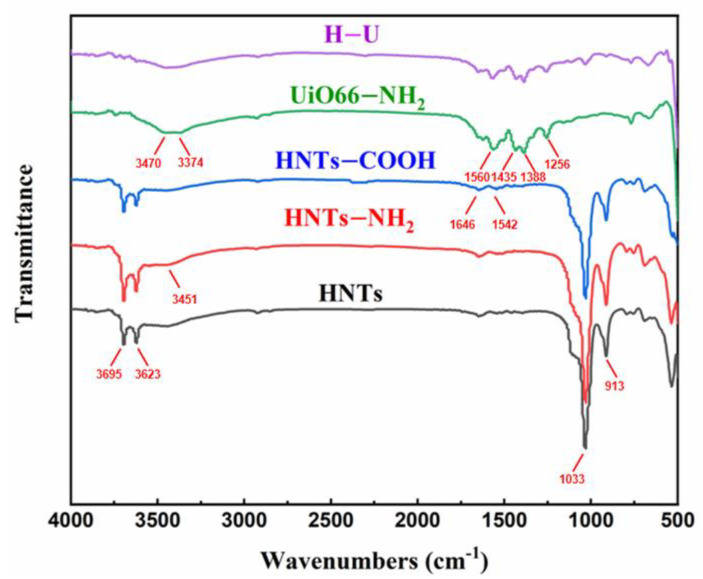
FT-IR spectra of HNTs, HNTs-NH_2_, HNTs-COOH, UiO66-NH_2_, H-U.

**Figure 4 polymers-13-03648-f004:**
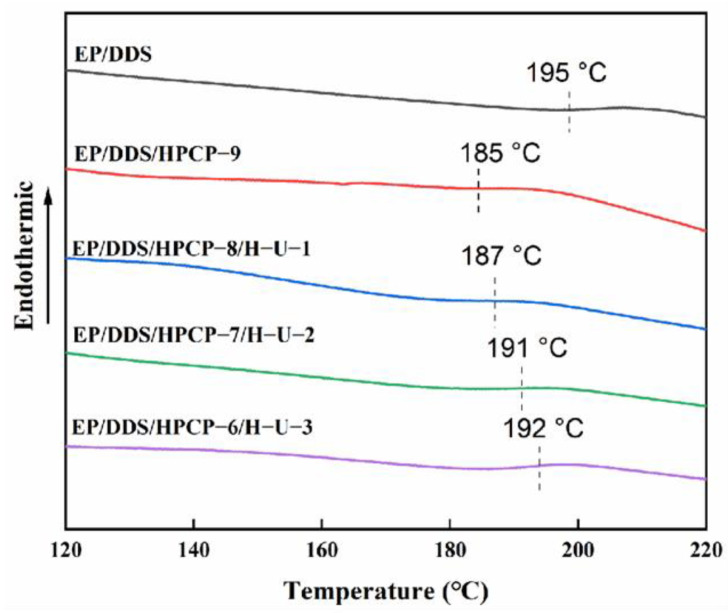
T_g_s of EP and its composites obtained by DSC.

**Figure 5 polymers-13-03648-f005:**
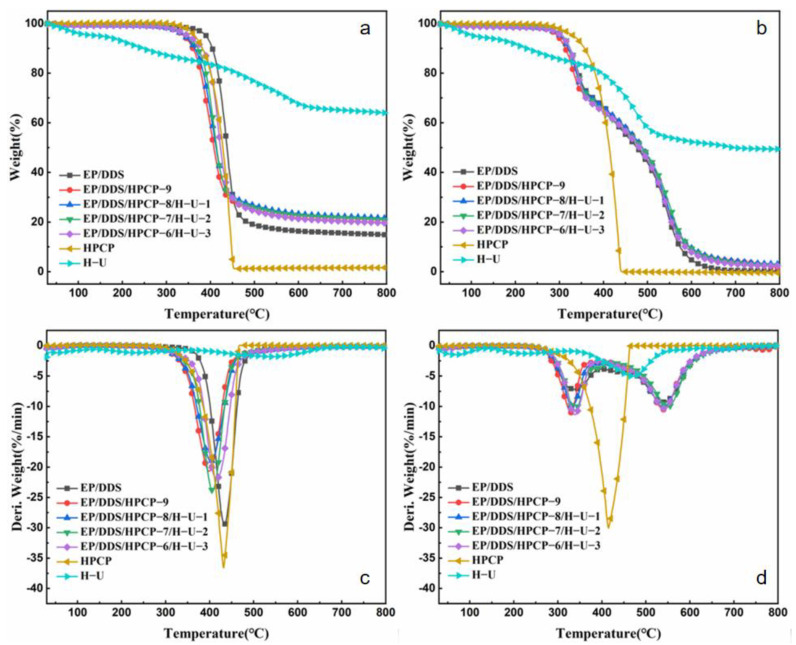
TG and DTG plots of EP samples, H-U and HPCP in N_2_ (**a**,**c**) and air (**b**,**d**) conditions.

**Figure 6 polymers-13-03648-f006:**
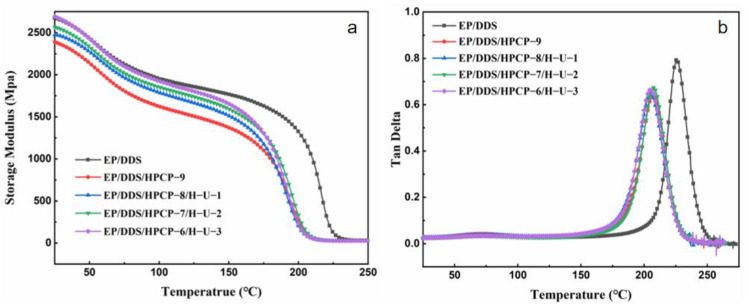
Storage modulus (**a**) and tan delta (**b**) curves of EP samples vs. temperature.

**Figure 7 polymers-13-03648-f007:**
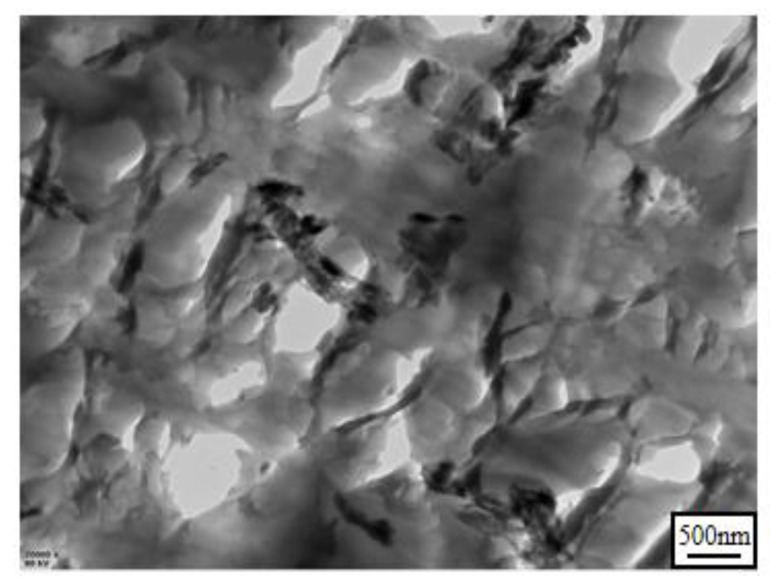
TEM image of the fracture surfaces of EP/DDS/HPCP-6/H-U-3.

**Figure 8 polymers-13-03648-f008:**
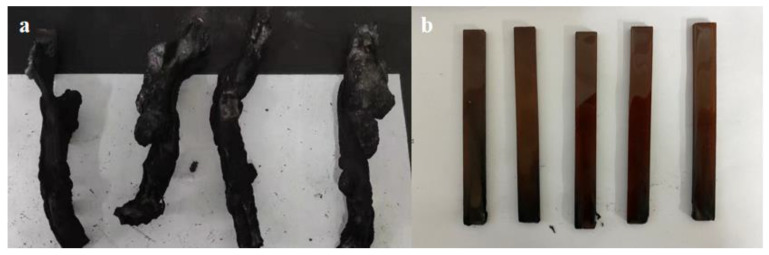
Digital images of EP/DDS/HPCP-9 (**a**) and EP/DDS/HPCP-6/H-U-3 (**b**) samples after UL-94 tests.

**Figure 9 polymers-13-03648-f009:**
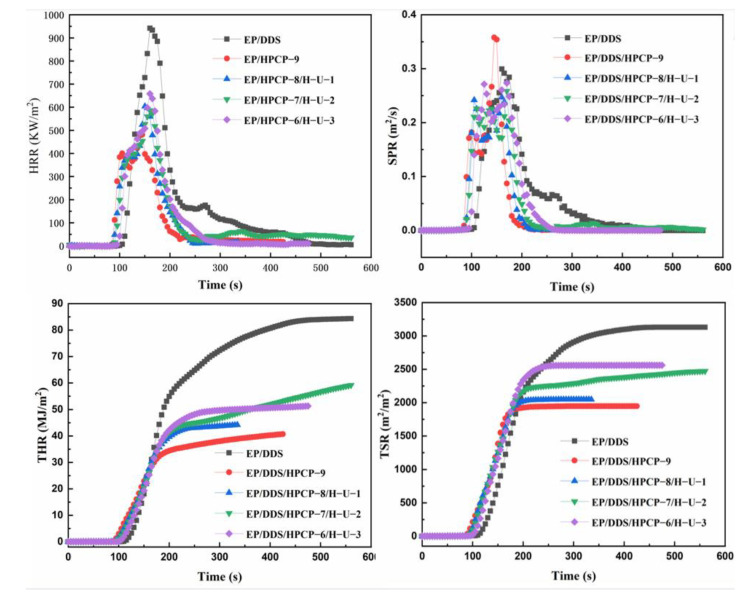
Plots of HRR (**a**), THR (**b**), SPR (**c**), and TSR (**d**) versus time of EP and its composites.

**Figure 10 polymers-13-03648-f010:**
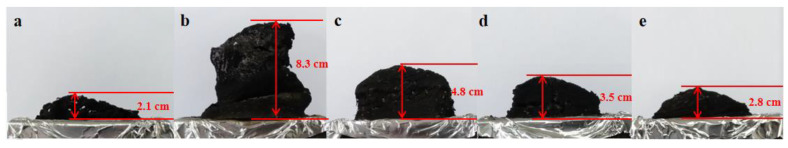
Digital photos of char layers of EP/DDS (**a**), EP/DDS/HPCP-9 (**b**), EP/DDS/HPCP-8/H-U-1 (**c**), EP/DDS/HPCP-7/H-U-2 (**d**), and EP/DDS/HPCP-6/H-U-3 (**e**) after cone tests.

**Figure 11 polymers-13-03648-f011:**
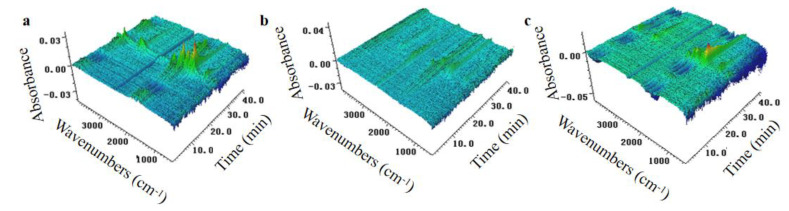
TG-IR spectra of thermal degradation products of EP/DDS (**a**), EP/DDS/HPCP/9 (**b**), and EP/DDS/HPCP-6/H-U-3 (**c**).

**Figure 12 polymers-13-03648-f012:**
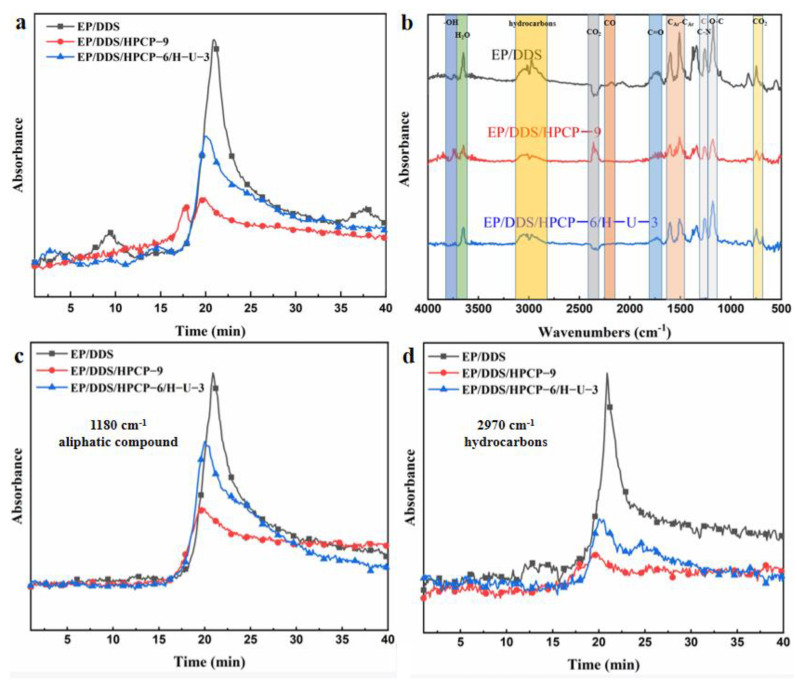
TG-IR spectra of decomposition products of EP composites: total absorbance (**a**), *T*_max_ (**b**), aliphatic compound (**c**), and hydrocarbons (**d**).

**Figure 13 polymers-13-03648-f013:**
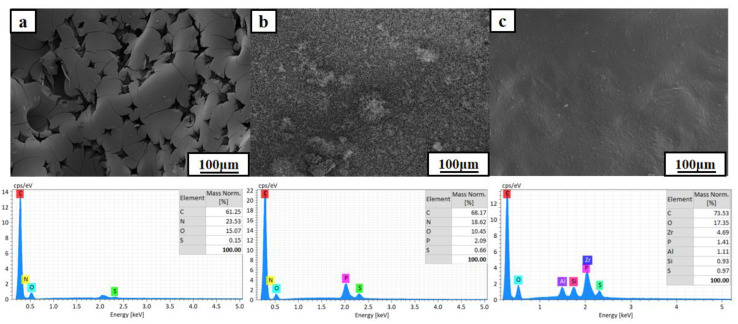
The SEM images and EDS spectra of char residues of EP/DDS (**a**), EP/DDS/HPCP-9 (**b**), and EP/DDS/HPCP-6/H-U-3 (**c**) after cone calorimeter tests.

**Table 1 polymers-13-03648-t001:** Formulas of EP and its composites.

Sample ID	EP/wt%	DDS/wt%	H-U/wt%	HPCP/wt%
EP/DDS	75.2	24.8	-	-
EP/DDS/HPCP-9	68.4	22.6	-	9.0
EP/DDS/HPCP-8/H-U-1	68.4	22.6	1.0	8.0
EP/DDS/HPCP-7/H-U-2	68.4	22.6	2.0	7.0
EP/DDS/HPCP-6/H-U-3	68.4	22.6	3.0	6.0

**Table 2 polymers-13-03648-t002:** TGA results of EP and its composites.

Atmosphere.	Sample	*T_5%_* (°C)	*T_max1_* (°C)	*R_max1_* (%/min)	*T_max1_* (°C)	*R_max2_* (%/min)	CY (%)
N_2_	EP/DDS	391	433	29.5	-	-	14.8
	EP/DDS/HPCP−9	339	399	20.7	-	-	20.0
	EP/DDS/HPCP-8/H−U−1	341	403	19.0	-	-	21.5
	EP/DDS/HPCP-7/H−U−2	348	406	23.8	-	-	20.8
	EP/DDS/HPCP-6/H−U−3	350	416	21.9	-	-	19.6
	HPCP	362	431	36.6	-	-	1.6
	H−U	147	538	1.8	-	-	63.9
Air	EP/DDS	306	338	7.2	537	9.3	0.3
	EP/DDS/HPCP−9	295	330	11.0	540	10.5	1.8
	EP/DDS/HPCP-8/H−U−1	302	332	9.9	539	10.0	3.0
	EP/DDS/HPCP-7/H−U−2	308	338	10.6	550	10.1	2.3
	EP/DDS/HPCP-6/H−U−3	306	339	11.3	542	10.3	2.3
	HPCP	332	415	30.1	-	-	0
	H−U	106	468	5.1	-	-	49.4

**Table 3 polymers-13-03648-t003:** Mechanical properties of EP thermosets.

Sample	Tensile Strength (MPa)	Elastic Modulus (MPa)	Flexural Strength (MPa)	Flexural Modulus (MPa)	*E′* (MPa)	*T_g_* (°C)
EP/DDS	71.2 ± 5.3	213 ± 100	109.0 ± 1.9	2932 ± 104	2642	225
EP/DDS/HPCP−9	58.6 ± 8.4	213 ± 71	91.5 ± 4.6	3124 ±143	2352	206
EP/DDS/HPCP−8/H−U−1	62.4 ± 2.4	281 ± 128	107.5 ± 2.8	3164 ± 107	2450	204
EP/DDS/HPCP−7/H−U−2	68.8 ± 7.5	343 ± 120	108.0 ± 5.0	2779 ± 120	2525	205
EP/DDS/HPCP−6/H−U−3	60.2 ± 2.1	514 ± 101	111.1 ± 6.6	3124 ± 115	2652	204

**Table 4 polymers-13-03648-t004:** Combustion data of EP samples.

Sample	LOI (%)	UL-94			TTI (s)	pHRR (kW/m^2^)	THR (MJ/m^2^)
		t_1_ (s)	t_2_ (s)	Rating			
EP/DDS	24.5	80	-	NR	100	942	84.4
EP/DDS/HPCP−9	28.4	-	-	NR	80	481	40.7
EP/DDS/HPCP−8/H−U−1	31.6	7	12	V-1	82	604	44.2
EP/DDS/HPCP−7/H−U−2	33.7	11	8	V-1	81	586	64.9
EP/DDS/HPCP−6/H−U−3	35.2	3	3	V-0	92	658	51.3

## Data Availability

All data is offered by corresponding author for reasonable request.

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
