# Peer review of "A Novel Synergistic Flame Retardant of Hexaphenoxycyclotriphosphazene for Epoxy Resin"

_polymers, 2021, doi:10.3390/polym13213648_

Round 1
Reviewer 1 Report
The manuscript by Jiang J. et al. describes on "A novel synergistic flame retardant of hexaphenoxycyclotriphosphazene for epoxy resin". They synthesized H-U by hydrothermal reaction by combining UiO66-NH2 (MOF) with HNT-COOH (halloysite nanotubes), which then added to HPCP for EP (epoxy polymers). The introduction of H-U can effectively improve the thermostability, tensile and flexural properties. The systems reached a V-0 rating and achieved a limited oxygen index of 35%. However, the height of char layer of the systems reduced due to the introduction of H-U that weakened their fire-retardant performances in cone calorimeter test. The fire-retardant properties are studied with several experimental techniques including TG-IR and TEM. Therefore, I recommend the contents of the manuscript suitable for publication in polymers. However, here is the comment that need to be addressed in the revised manuscript for its publication.
In the references section, there are several references that are incomplete as written. The references are 13, 23, 25, 28, 44 and 45 that need to be completed. Please go through references carefully for correctness.
Author Response
Response to Reviewer1
Comment 1: In the references section, there are several references that are incomplete as written. The references are 13, 23, 25, 28, 44 and 45 that need to be completed. Please go through references carefully for correctness.
Response: Thank you for your advice. We have been checked and revised the references.

Reviewer 2 Report
Tha manuscript is good enougn for publishing; although there are some points to be corrected:
The pdf file presents many typos, probably due to the platform, but there should be corrected
The introduction should be improved describing more antecedents about the application of MOF, describing the previous knowledge (the theory behind that explain how these materials improve the FR) and the lagoon; and also how the authors use that information to produce new knowledge
Figure 4 is a bit confusing; it is not as clear as the text; please clarify the graphical description
Carbonyl vibration values in the text are not the same that in the FTIR figure
Author Response
Response to Reviewer2
Comment 1: The pdf file presents many typos, probably due to the platform, but there should be corrected.
Response: Thank you for your advice. We have been checked the whole text and revised the grammar and spelling mistakes as much as possible.
Comment 2: The introduction should be improved describing more antecedents about the application of MOF, describing the previous knowledge (the theory behind that explain how these materials improve the FR) and the lagoon; and also how the authors use that information to produce new knowledge
Response: Thank you for your advice. More detailed instructions that how MOF improve the flame retardancy for materials have been added. we analyzed these mechanisms so we tried to apply them to epoxy resins .
Comment 3: Figure 4 is a bit confusing; it is not as clear as the text; please clarify the graphical description.
Response: Thank you for your advice. The caption of Figure 4 has been revised as Tgs of EP and its composites obtained by DSC.”
Comment 4: Carbonyl vibration values in the text are not the same that in the FTIR figure.
Response: Thank you for your advice. The sentence has been revised as “The carbonyl vibration peaks of HNTs-COOH could be observed at 1542 and 1646 cm-1, respectively”, which is the same as the figure.
